# Faith, Knowledge, and the *Ausgang* of Classical German Philosophy: Jacobi, Hegel, Feuerbach

Todd Gooch 

Department of History, Philosophy, and Religious Studies, Eastern Kentucky University, Richmond, KY 40422, USA; todd.gooch@eku.edu

**Abstract:** This article revisits Feuerbach's "break with speculation" in the early 1840s in light of issues raised by the original Pantheism Controversy, initiated in 1785 by the publication of Friedrich Heinrich Jacobi's *Letters on the Doctrine of Spinoza*. The article first describes the concerns underlying Jacobi's repudiation of Spinozism, and rationalism more generally, in favor of a personalistic theism that disclaims the possibility of philosophical knowledge of God. It goes on to reconstruct Hegel's alternative to Jacobi's famous *salto mortale* before considering how Feuerbach's critique of Hegel's philosophy of religion, as well as the personalism of the so-called Positive Philosophy (inspired by the late Schelling), was influenced by both Spinoza and Jacobi in ways that have not yet received sufficient attention.

**Keywords:** Pantheism Controversy; German Idealism; philosophy of religion; F.H. Jacobi; G.W.F. Hegel; Ludwig Feuerbach

## 1. Introduction

The publication of Ludwig Feuerbach's *The Essence of Christianity* in 1841, followed in 1843 by his *Principles of the Philosophy of the Future*, has sometimes been taken to mark the end or *Ausgang* of the period in the history of classical German philosophy that began with the appearance of Kant's *Critique of Pure Reason* in 1781.[1] This article revisits Feuerbach's "break with speculation" in light of issues raised by the critique of philosophical rationalism advanced by Friedrich Heinrich Jacobi, initially in his letters to Moses Mendelssohn *On the Doctrine of Spinoza*, first published in 1785. In doing so, it seeks to cast in a new light certain debates about religion that are closely associated with the decline of German Idealism during the middle of the nineteenth century.

Although no match for Kant in terms of the breadth of his philosophical vision or the thoroughness of its execution, probably no one other than Kant exercised a greater influence on the development of classical German philosophy than did Jacobi. This influence was not limited to the debates about the relationship between religious faith and scientific (i.e., *wissenschaftlich*) knowledge that are the focus of this article. It extended to fundamental issues of philosophical method reflected in the various attempts of Fichte, Schelling, and Hegel to develop a "system of reason" capable of responding both to Jacobi's critique of Kant's transcendental idealism and to his insistence upon the fundamental irreconcilability of a thorough-going philosophical rationalism with a belief in human freedom (cf. Jaeschke and Arndt 2012, esp. pp. 23–37 and 131–61; Sandkaulen 2023).

Jacobi's philosophical influence on these and other thinkers occurred primarily through his instigation of, and/or participation in, three public controversies (*Streiten*), which many of the most eminent German thinkers of his age weighed in on. These controversies raised issues that continued to shape debates among various proponents of pantheism and personalism for several decades, both in Germany and in Great Britain (cf. Bengtsson 2006). They include (1) the original Pantheism Controversy inaugurated by the publication in 1785 of Jacobi's Spinoza Letters, (2) the Atheism Controversy that resulted in Fichte's dismissal

from his chair at Jena in 1799, and (3) the exchange between Jacobi and Schelling initiated by the former through the publication in 1811 of his essay, "On the Divine Things and their Revelation" (cf. Essen and Danz 2012; Timm 1974).

Jacobi's enduring influence on Hegel is reflected in the frequency with which explicit and implicit references to his ideas occur throughout Hegel's published works and lectures, including the 1802 essay on *Faith and Knowledge*, the *Science of Logic*, the various editions of the *Encyclopedia of the Philosophical Sciences*, the 1822 foreword to Hinrichs' *Religion in Its Inner Relation to Science*, as well as the lectures on the philosophy of religion. The synthesis of faith and knowledge proposed by Hegel in these works, which involves a novel reconceptualization of both reason and revelation (described below), was developed partly in response to Jacobi (and partly to a range of thinkers including the Berlin *Aufklärer*, Kant, Fichte, Schleiermacher, and such neo-Pietists as August Tholuck, among others).

A considerable part of the appeal of the Hegelian philosophy to many of those who embraced it in the 1820s and 1830s was the hope it seemed to offer for reconciling their commitment to the Christian faith with their commitment to the freedom of scientific inquiry. Nevertheless, what D. F. Strauss referred to as the "beautiful hope-filled days of peace for theology" that the Hegelian synthesis of faith and knowledge seemed at first to presage were short-lived (Strauss 1840, p. 1).[2] Debates about the compatibility of Hegel's philosophy with the doctrines of the state-sanctioned ecclesiastical bodies became increasingly heated toward the end of the 1830s. These debates, together with political developments related to them, led to the rapid fragmentation, and eventual dissolution, of the Hegelian school.

One of the most vocal participants in these debates was Feuerbach, who had attended Hegel's lectures in Berlin from 1824 to 1826 and defended the Hegelian cause skillfully in essays and reviews published in the 1830s before beginning to distance himself from that cause in print in 1839. The publication, in 1841, of Feuerbach's magnum opus, *The Essence of Christianity*, marks a further step in this direction, though it was not until 1843 that a decisive break was announced publicly with the appearance of Feuerbach's "Preliminary Theses for the Reformation of Philosophy", followed shortly thereafter by his *Principles of the Philosophy of the Future*.[3] Although Jacobi's influence on Hegel is widely acknowledged, his influence on Feuerbach has, until recently, gone largely unnoticed.[4] Nevertheless, as Christine Weckwerth, one of the co-editors of the critical edition of Feuerbach's collected works, has suggested, "Feuerbach's polemic against monotheism can . . . be viewed as a late offshoot of the Spinoza Controversy, which thereby enters into the discussion of early Hegelianism and the nascent philosophy of the *Vormärz*" (Weckwerth 2004, p. 433).[5]

My aim here is to take up this suggestion by focusing on the use made by Feuerbach of ideas taken over from both Spinoza and Jacobi in conceptualizing the break with the speculative philosophical tradition of which he had himself been a vocal advocate during the 1830s. In the *Principles of the Philosophy of the Future*, Feuerbach claimed for his "new" sensualistic philosophy that it is the incarnational telos of the historical development of modern philosophy ("*das fleisch-und-blut-gewordene Resultat der bisherige Philosophie*"), a topic on which Feuerbach had lectured and published extensively during the 1830s. The more modest thesis I seek to defend here is that there is a meaningful sense in which Feuerbach's philosophy of the future—which he enigmatically concedes at one point is "no philosophy" at all ("*keine Philosophie ist*")—is a historical heir of Jacobi's "*Unphilosophie*". To that extent, what Engels referred to as the "*Ausgang*" of classical German philosophy seems to be more closely related to its *Eingang* that has generally been recognized.

In defending this thesis, I shall revisit Feuerbach's break with speculation in light of philosophical issues raised by the original *Pantheismusstreit*. The discussion proceeds in three parts. In the first part, I describe the concerns underlying Jacobi's repudiation of philosophical rationalism in favor of a personalistic theism that disclaims the possibility of philosophical knowledge (*Wissen*) of God. In the second part, I describe the reasons for Hegel's dissatisfaction with Jacobi's account of the relationship between faith and knowledge, as well as the strategy underlying his alternative proposal (cf. also Stewart

2018). In the third part, I attempt to show how Feuerbach's critique of the Hegelian claim for the identity of religious and philosophical truth, as well as his critique of the so-called "Positive Philosophy" inspired by the late Schelling, were influenced by both Jacobi and Spinoza in crucial ways that have yet to receive sufficient consideration.

I hope, in drawing attention to these influences, to contribute to a reassessment of Feuerbach's role in the history of German Idealism and its aftermath, which saw the emergence of several developments that can be said to have been influenced by Feuerbach, either directly or indirectly. These include the rise of scientific materialism and positivism, several distinct streams of socialist thought, drive psychology, and *Existenzphilosophie* with its characteristic concerns for facticity, temporality, corporeality, and finitude. It is presumably with developments like these in mind that Karl Löwith could write, with some plausibility, in 1964 that "Feuerbach's effort to make Hegel's philosophical theology tangible and finite simply became the standpoint of the age. Now—consciously or unconsciously—it belongs to all of us" (Löwith 1964, p. 82).

## 2. Jacobi

The history of the influence (*Wirkungsgeschichte*) of Jacobi's Spinoza Letters is largely, though not entirely, a history of unintended consequences. According to the account of his fateful conversation with Lessing contained in them, Jacobi had come to Wolfenbüttel in 1780 with the intention of enlisting Lessing's support "against Spinoza". He was taken aback when, after reading Goethe's poem fragment, "Prometheus", Lessing, without knowing the identity of the author, expressed his approval of the standpoint expressed in it. In doing so, he declared his dissatisfaction with "the orthodox concepts of the Divinity", as well as his commitment to a view of God as *hen kai pan* or One and All (Jacobi 1994, p. 187).[6]

In 1789, Jacobi introduced his contemporaries to the heterodox ideas of Giordano Bruno by including in the first appendix to the second edition of the Spinoza Letters excerpts from Bruno's *De la causa, principio e uno* (1584), a work he claimed belongs, together with Spinoza's *Ethics*, to "the *Summa of the philosophy of the* Hen kai Pan" (emphasis in original). Though it was not his intention to do so, as di Giovanni has observed, he thereby contributed "one more source of inspiration for the tendency to divinize nature already at work in the incipient Romantic movement" (Jacobi 1994, p. 192). Jacobi could scarcely have imagined that, within a few years, Spinoza would be hailed by Novalis and Schleiermacher as a God-intoxicated sage; that the first articulation of the so-called identity philosophy would be presented in the form of a dialogue named by Schelling for Bruno; or that, three decades later, Ludwig Feuerbach, the brother of Jacobi's own *Patenkind*, would be proud to be have been dubbed "Bruno reincarnate" by one of his friends (cf. Gooch 2013a).

Far from being intended as an endorsement of Spinozism, Jacobi's Spinoza Letters are an expression of existential protest against what he considered to be the fatalistic implications of a consistent philosophical rationalism, of which he took Spinoza's philosophy to be an unsurpassed exemplar. That said, Jacobi's stance toward Spinoza is complex and ambivalent (cf. Sandkaulen 2023, pp. 15–27). "I love Spinoza", he remarks to Lessing at one point in the Letters, "because he, more than any other philosopher, has led me to the perfect conviction that certain things admit of no explication: one must not therefore keep one's eyes shut to them, but must take them as one finds them" (Jacobi 1994, p. 193).

It is fairly common knowledge that Jacobi proposed to extricate himself from the fatalistic implications Spinoza's philosophy through his famous *salto mortale* (in its literal sense, a type of acrobatic somersault). Statements like the following have often been interpreted as expressing a leap of religious faith that has given rise to charges of obscurantism and religious irrationalism: "The whole thing comes down to this: from fatalism I immediately conclude against fatalism and everything connected with it" (Jacobi 1994, p. 189). In fact, this remark merely encapsulates a sophisticated epistemological argument against the kind of systematic rationalism epitomized by Spinoza that is similar in some respects to the proof of an external world put forward in the twentieth century by G. E. Moore (cf. Moore 1939). Rather than being an expression of irrationalism, Jacobi's leap involves an appeal

to a kind of unmediated certainty that trumps the degree of certainty of any conclusion derived by means of an inference from antecedent premises (cf. Crowe 2009; Beiser 1987, pp. 44–126). Jacobi regards his leap as warranted by the consideration that "I have no concept more intimate than that of final cause; no conviction more vital than that *I do what I think*, and not, *that I should think what I do*" (Jacobi 1994, p. 193; emphasis in the original). Since no conclusion reached through demonstrative reasoning can produce a degree of certainty as great as that with which he holds this conviction, Jacobi contends that he is in his rights to reject any such conclusion out of hand.

When asked by Lessing to share what he took to be the "the spirit that inspired Spinoza himself", Jacobi replied: "It is certainly nothing other than the ancient *a nihilo nihil fit*" (Jacobi 1994, p. 187). The cornerstone of Spinoza's philosophy is his definition of substance as "that which is in itself and is conceived through itself" (de Spinoza 1992, p. 31), and which he demonstrates from this definition to be necessarily one and infinite. Operating with this conception, Jacobi explains, Spinoza "established that with each and every coming-to-be in the infinite . . . something is posited out of nothing" (Jacobi 1994, pp. 187–88). The evident impossibility of this led Spinoza to reject the monotheistic conception of God as the transient cause of nature in favor of a conception of Deus sive natura (i.e., God or Nature) as the "immanent infinite cause", which is "One and the same with all its consequences" (ibid., p. 188). Possessing "neither understanding nor will", this immanent cause acts without "intentions and final causes" (ibid.) while also being devoid of affect and therefore incapable of love. The ordinary belief that we are responsible for our actions Spinoza attributes to ignorance of the true causes of our desires and volitions (de Spinoza 1992, p. 57).

Jacobi's protest against the fatalistic implications of Spinoza's philosophy, which he regarded as irrefutable on its own terms, and his rejection of attempts like Herder's to produce a "refined" (*geläuterte*) version of Spinozism that might avoid these implications, are registered on behalf of what he refers to in his open letter to Fichte (1799) as "the highest in man" (Jacobi 1994, p. 499). Jacobi seems to associate the unique dignity of human beings with feelings of admiration, respect, and love. These are only properly directed toward persons in whose wills originate achievements that are deserving of the admiration we sometimes feel toward them. If there are no final causes, then, strictly speaking, no one is worthy of admiration, respect, or love since no one is the originator of their own thoughts and actions. Indeed, in that case, "the only function that the faculty of thought has in the whole of nature is that of passive observer; its proper business is to accompany the mechanism of efficient causes" (Jacobi 1994, p. 189). It is this conclusion that Jacobi seeks to repudiate by means of an appeal to the immediate certainty with which he is aware that he is the cause of his own actions, even though he acknowledges that he lacks the resources to demonstrate this claim and must, in affirming it, "assume a source of thought and action that remains entirely inexplicable to me" (Jacobi 1994, p. 193).

To appreciate Jacobi's position, it is important to recognize how the affirmation of human freedom, belief (*Glaube*) in the existence of a personal Creator, and denial of the possibility of rational cognition of God are logically interrelated in his way of thinking. Jacobi's admission that he lacks the resources for explaining either how free will is possible, or how it is possible for God to have created the universe from nothing, is due to his conviction that freedom is by its very nature inexplicable. This is the case insofar as any effort to explain how it is possible for a human volition to produce a human action, or a divine volition to produce a divine action, would subsume volition within the system of efficient causes. For Jacobi, whatever is scientifically explicable is by definition natural. "Everything that reason can produce through division, combination, judgment, inference, and reflection is simply a natural thing" (Jacobi 1994, p. 192).[7] That human beings exist physically as part of a deterministic system of natural causes, while simultaneously possessing the capacity for self-determination, Jacobi takes to be "an absolutely incomprehensible fact; a *miracle* and a *mystery* comparable to creation" (Jacobi 1994, p. 530).[8]

While Jacobi claims indubitability for his beliefs both that he is the cause of his own actions and that there exists a personal God who created the world ex nihilo—not to mention

his belief in the existence of an extra-mental reality that is revealed to him by the senses—the kind of certainty involved here is not the same kind of certainty produced by rational demonstration. In his open letter to Fichte, Jacobi remarks somewhat cryptically that he associates "the highest in man" with "consciousness of non-knowledge [*des Nichtwissens*]" and that he identifies the "location" (*Ort*) of this consciousness with "the location of *the true* that is inaccessible to *Wissenschaft*" (Jacobi 1994, p. 499). *Nichtwissen* or non-knowledge, in the specific sense in which Jacobi employs this term, is thus not synonymous with ignorance but refers instead to the faith (*Glaube*) or immediate cognition through which Jacobi contends that supernatural truths are apprehended by the human mind. Freedom and non-knowledge, understood in this way, are co-extensive (cf. Jacobi 1994, p. 532).

The truth to which *Nichtwissen* gives access is a revealed truth rather than a demonstrable one. It is not scripture, however, that is the primary vehicle of this revelation but rather the individual human personality and the moral goodness that it is uniquely capable of manifesting in the world. Insofar as every "purely ethical, truly virtuous, action . . . [is] a miracle with respect to nature", each such action "reveals *Him* who only can *do* miracles, the creator, the almighty Lord of nature, the ruler of the universe" (Jacobi 1994, p. 589).[9] It is on the basis of this understanding of the relation between freedom and non-knowledge that Jacobi identifies the "jewel" of the human race, not with a "science that does away with all miracles", but instead with "the faith in a Being who can only do miracles, and who also created man miraculously; the faith in God, freedom, virtue, and immortality" (Jacobi 1994, p. 561).

## 3. Hegel

Hegel agrees with Jacobi that the method of rational demonstration employed by the reflective understanding is not suitable for obtaining philosophical knowledge of God or the Absolute. However, he does not share Jacobi's assumption that the kind of cognition of finite objects involved in "secular" science is the only type of cognition there is.[10] Against Jacobi (and others who discount the possibility of God's being rationally comprehensible by human beings due to their epistemic limitations), Hegel insists that the God of Christianity is not one who "enviously" hides himself from His creation, but rather one whose very nature it is to reveal himself to the community of finite or subjective spirits fashioned in His image (to employ the language of religious representation). The content of the Christian revelation thus demands to be comprehended conceptually. This, however, is not a task for which the reflective understanding (*Verstand*) that we employ in the cognition of finite objects is suited. Only speculative reason (*Vernunft*) as conceived by Hegel in his *Science of Logic* is capable of comprehending discursively the determinations of the absolute Idea.

An important impetus for the new logical method that Hegel sought to develop in his own *Science of Logic* was Kant's discussion of the antinomies of reason in the First Critique. Hegel credited Kant with having shown dialectic to be "a necessary function of reason" and having "vindicated . . . the objectivity of the illusion and the necessity of the contradiction which belongs to the nature of thought determinations" (Hegel 1969, p. 56). Thus, the arrival of the time—at the end of the *Aufklärung*—when it had finally become possible for philosophy "to deal with religion more impartially, on the one hand, and more fruitfully and auspiciously on the other", was signaled for him by the appearance of a new approach to logic first made possible by this Kantian breakthrough.

In his lectures on the philosophy of religion, Hegel seeks to develop speculatively the concept of religion and thereby to show that this concept reaches its complete development in Christianity, which deserves for this reason to be called the "*vollendete*" or consummate religion. In doing so, he makes the following methodological remark: "There can be but one method in all science, in all knowledge. Method is just the self-explicating concept—nothing else—and the concept is one only" (Hegel 1988, p. 100).[11] This goes to show that Hegel conceives of logic as the science of that Concept from which the form of all individual concepts, including the concept of religion, is derived (Hegel 1969, p. 30). Each of these is "a moment of the form as totality of that same Concept which is the foundation of the

specific concepts" (Hegel 1969, p. 39). The Concept with which Hegel's Logic is concerned is the exclusive product and object of thinking and is nothing other than "the logos, the reason of that which is, the truth of what we call things" (Hegel 1969).

In his 1822 foreword to Hinrichs' *Religion in Its Inner Relation to Science*, which Westphal calls "the most lucid and succinct statement of Hegel's mature position on the question of faith and reason", Hegel affirms the rightful claims of both (Hegel 2002, p. 332). In doing so, he argues that any satisfactory reconciliation between the two must be one that avoids the Scylla of depriving faith of its substantial content (so that "only the empty husk of subjective conviction remains"), as well as the Charybdis of depriving reason of its theoretical ambition to discover the truth (Hegel 2002). What seems to count for Hegel as "the highest in man" is not, as for Jacobi, faith in a personal creator whose nature defies conceptual comprehension; it is rather the thinking activity in which the reality of spirit consists. The aim of this activity is the comprehension of the content of the concrete Concept in the fullness of its internal determinations. Christian faith, in what Hegel considers to be its true sense, encompasses both the subjective element of unshakable conviction *and* the objective element of creedal content. Religion, on this view, "has its seat and soil in the activity of thinking", so that, even when "the truth of religion is . . . directly sensed [by means of] the heart and feeling", these remain "the heart and feeing of a thinking human being" (Hegel 1988, p. 399). That is, even where the doctrinal contents of the faith have not yet been rationally comprehended, genuine faith involves the possibility of their becoming so—a possibility for which Jacobi's account of faith as non-knowing does not allow.

In Hegel's view, spirit is free to the extent that it has come to be for itself. To that extent, spirit transcends nature while at the same time presupposing it. What needs here to be emphasized is that the freedom in which spirit is at home with itself, and in which it essentially consists, is one that is achieved only gradually through its own labors. These include the history of the religious representations that it has produced as it has struggled toward self-consciousness. This freedom is not accomplished in a single leap, much less one that it is in the power of any individual to make. It is achieved instead in the course of centuries through a collective agency (exercised by an "I that is We") in which the individual thinking subject participates, but only to the extent that he or she succeeds in transcending the contingency of his or her own subjective preferences and opinions.

Whereas Hegel disagrees with Jacobi in maintaining that God can be an object of rational cognition, his disagreement with Spinoza is expressed in his claim that "the standpoint of the Concept" is a higher standpoint than that of Essence, containing and preserving the truth of the latter while at the same time superseding it, as well as in his identification of the logical doctrine of Essence (*Wesen*) with necessity, and the doctrine of the Concept (*Begriff*) with freedom and self-consciousness. In the *Encyclopedia* Logic, Hegel affirms that "the various stages of the logical Idea can be considered as a series of definitions of the Absolute" (Hegel 1991, p. 237 [§160 Z]). In the first part of the Logic, on the doctrine of Being (*Sein*), which Hegel refers to as "the Concept only *in-itself*", the Absolute is defined as Being (Hegel 1991, p. 135 [§84] and p. 137 [§86 Z]). In the second part, on the doctrine of Essence, which is equated at one point with "the Concept as *posited* Concept" (Hegel 1991, p. 175 [§112]), the Absolute is defined not as a highest essence that is "'given' while outside and beside him there are also other essences", but rather as the *infinite* essence within which all finite essences and their internal conceptual relations are contained (Hegel 1991, p. 177 [§112 Z]). The "objective logic", which encompasses the doctrines of Being and of Essence taken together, is said by Hegel to take the place of the former metaphysics.

In the transition from the "objective" to the "subjective" logic (or doctrine of the Concept) in the *Science of Logic*, Hegel acknowledges that Spinoza's idea of substance is "a *necessary standpoint* assumed by the absolute" but claims that it is not yet "*the highest standpoint*" (Hegel 1969, p. 580; emphasis in the original). In a related remark that seems to be directed against Jacobi's *salto mortale* as a response to Spinoza, he writes: "The genuine refutation must penetrate the opponent's stronghold and meet him on his own ground;



no advantage is gained by attacking him somewhere else and defeating him where he is not" (Hegel 1969, p. 581). That is, a genuine refutation of Spinoza must show how the standpoint of Essence, through its own internal contradictions, itself gives rise to the higher standpoint of the Concept. It is precisely in its failure to raise itself to the standpoint of the Concept that Spinoza's philosophy is said to have fallen "short of the true concept of God which forms the content of the Christian religious consciousness" (Hegel 1991, p. 226 [§ 151 Z]). It is to their failure to appreciate the difference between these two standpoints that Hegel attributes the misguided charge of pantheism leveled against his position by critics such as the Pietist theologian, Tholuck.

To conceive of God according to the standpoint of the Concept is, on this account, to do something that Spinoza could not bring himself to do, namely, to posit finitude within God himself, only "not as something insurmountable, absolute, independent, but above all as [a] process of distinguishing that[,] . . . because it is a transitory moment and because finitude is no truth, is also eternally self-sublating" (Hegel 1988, p. 406).[12] It is in the form of the Christian doctrine of the Incarnation that this "positing" first occurs, and God is thereby first conceived as spirit. This involves the pre-existent divine Logos being "transplanted into the world of time, . . . putting himself in judgment and expiring in the pain of negativity", even while, "as infinite subjectivity, keep[ing] himself unchanged" (Hegel 1971, p. 300 [§ 569]). In the Christian doctrines of the Trinity and the Incarnation, the absolute self-mediation of spirit is "set out as a cycle of concrete shapes in pictorial thought" (Hegel 1971, p. 301 [§ 571]). Nevertheless, the comprehension of the positive content of these revealed truths that is achieved by philosophical thinking does not rest on an appeal to the authority of the biblical texts or the facticity of events narrated in them. This is because philosophy is not meant, for Hegel, "to be a narration of happenings but a cognition of what is true in them, and further, on the basis of this cognition, to comprehend that which in the narrative appears as a mere happening" (Hegel 1969, p. 588). To grasp the truth of the Christian faith in this way is to comprehend it conceptually and thereby to recognize its inherent rationality.

The doctrines of the Christian religion as they are presented in the Bible and in the creeds and catechisms of the Church are "given in a positive fashion" (Hegel 1988, p. 399). The form in which they are initially affirmed is the form of immediate certainty. This affirmation involves an act of faith wherein the spirit of the believer testifies to the truth content of the Christian revelation. Faith and reason are reconciled, however, only insofar as the positive truths thus affirmed are recognized, through an act of philosophical comprehension, to be necessary and eternal truths of reason. On this view, reason and revelation are *not* two categorically distinct sources of knowledge. The truths of revelation are rational truths. They are recognized as such once the truth content contained in them is transformed speculatively into a form that is adequate to itself. The spirit that reveals itself in the form of positive revelation and the thinking spirit that appropriates the contents of revelation in the form of the Concept are one and the same spirit. It is only through this process of self-mediation, Hegel maintains, that "the concept on its own account liberates itself truly and thoroughly from the positive" (Hegel 1988, p. 402).

Hegel's alternative to Jacobi's *salto mortale* is thus to acknowledge the truth of Spinoza's position while seeking at the same time to demonstrate that Spinoza fails to achieve "the highest standpoint", which is the one set forth in Hegel's own doctrine of the Concept, and which, as noted previously, he identifies with "the content of the Christian religious consciousness". According to that doctrine:

> Freedom is the following aspect of the idea: the concept, conceptually at home with itself, is free. The idea alone is what is true, but equally so is freedom. The idea is what is true, and the true is absolute spirit. This is the true definition of spirit. (Hegel 1988, p. 412)

In responding thus to Spinoza, Hegel takes himself to have shown, contra Jacobi, that a deductive system of reason "*aus einem Stuck*" is after all compatible with belief in freedom.

Whether Jacobi would have been satisfied with the definition of freedom produced by Hegel in this passage is another matter.

## 4. Feuerbach

That Feuerbach is generally (and rightly) remembered as an atheist and a materialist has tended to obscure the fact that he began his philosophical career as an enthusiastic pantheist. In his first book, *Thoughts on Death and Immortality* (1830), Feuerbach used the resources of speculative logic he had acquired from Hegel to develop a conception of the Divinity as One and All along lines laid out by Spinoza, Bruno, and Jacob Boehme (cf. Gooch 2013a). In light of its previously noted role in Jacobi's Spinoza Letters, it is telling that Feuerbach chose as an epigram to this book verses from Goethe's poem fragment, "Prometheus", whereas after his "break with speculation" (in 1842), Feuerbach would assign to the "philosophy of the future" (in a passage difficult to render in English) the task of pulling philosophy down, "*aus der göttlichen, nichtsbedürfenden Gedankenseeligkeit in das menschliche Elend*", in this earlier work, he instead regarded the contemplation of infinite substance as the highest ethical act of which human beings are capable (Feuerbach 1970a, p. 264). He also clearly shared Hegel's estimation of the "vanity" of the culture of reflection, which had forgotten the oneness, universality and infinity of reason (the topic of Feuerbach's doctoral dissertation) as the principal obstacle preventing modern subjects from achieving this, their highest, good.[13]

In addition to the early Feuerbach's evident attraction to the monistic conception of Divinity affirmed by Lessing in the Spinoza Letters, further evidence of his engagement with Jacobi in the 1830s is found in a review of a work entitled *Jacobi and the Philosophy of His Age* by J. Kuhn that Feuerbach contributed to the *Berlin Annals* in 1834 (Feuerbach 1969b). While conceding to Jacobi that Descartes' adoption of the method of mathematical demonstration was not entirely salubrious in its consequences for the development of early modern philosophy, Feuerbach objects in this review to Jacobi's account of the role of demonstration in Cartesian rationalism, specifically with respect to Jacobi's designation of the knowledge of God resulting from such demonstration as "mediated". Cartesian knowledge of God, who is imperceivable by nature, is the result of an act of thinking (*Denken*), and thinking, as conceived here by Feuerbach, involves abstracting from what is given to the senses. To be sure, the demonstration that God's existence is part of God's essence consists of the sequential, and hence temporal, presentation of a series of reciprocally conditioned propositions. Nevertheless, Feuerbach contends, the cognition in which this demonstration culminates is itself the *thought of* the timeless and "immediate" identity of God's essence and existence.

Two of the essays and reviews that Feuerbach contributed in the 1830s, first to the *Berlin* and later to the *Halle Annals*, were directed against authors identified by him as representatives of the so-called "Positive Philosophy". These thinkers, who include Friedrich Julius Stahl and Jacob Sengler, followed the lead of the late Schelling in their efforts to establish the freedom and personality of God as their fundamental *Grundsatz* or principle. They did this in explicit opposition to Hegel's absolute Idea and alleged *Begriffspantheismus*, as well as to philosophical rationalism more generally, which they thought had reduced God to a mere concept and compromised his sovereignty by subjecting his nature to rational necessity.[14] Both Stahl and Sengler had attended lectures on the history of modern philosophy delivered by Schelling in Munich in the late 1820s. In these lectures, Schelling had argued that the knowledge acquired by "negative" philosophy through the dialectical unfolding of the Concept completes itself only when it finally succeeds in distinguishing "*that to which* it is substance or is subordinated [i.e., God] . . . from *itself*". This, he claimed, occurs first as the result of "a subjective act, roughly comparable to the act of worship", whereby the merely negative logical method employed by Hegel "destroys itself in faith [*Glaube*], but precisely thereby posits what is truly positive and divine" (von Schelling 1994, pp. 175–76).[15]



What Schelling refers to here as that which is "truly positive and divine" is what the Positive Philosophers themselves affirm, according to Feuerbach, as the "highest essential concept and principle of [their] theological speculation", namely, "God as a personal being or the absolute personality" (Feuerbach 1969c, p. 183). For Feuerbach, who at this stage still identifies science (*Wissenshaft*) with thinking (*Denken*) conceived in Hegelian terms, this involves a fundamental confusion. Precisely because the individual personality (what Feuerbach calls the personality "as *concretum*") cannot be subsumed under any concept (by virtue of its being "*das von mir Unabsonderliche an mir*") and is for this reason quite literally incomprehensible, it is not a suitable object of "speculation" (as a method of philosophical inquiry).

On this reading, in seeking to make the individual personality the foundational principle of philosophical science (and thereby to avoid an inevitable and discomfiting "either/or"), the Positive philosophers remain true neither to the Christian faith nor to philosophical science. Although Feuerbach considered Jacobi's philosophy to be "a self-annihilating philosophy" because it "[puts] imaginary thought [*die* Einbildung, *zu denken*] in place of real thought" (Feuerbach 1984, p. 122), this did not prevent him from acknowledging Jacobi as "a classical, because consistent, philosopher [who was] at one with himself" in a way he did not think true of either the late Schelling or his disciples. This is because Jacobi never lost sight of the fact that "the personality proves itself only in a way that is itself personal" (Feuerbach 1973, pp. 188–89). This allowed him to preserve a clear distinction between science and non-knowledge (*Nichtwissen*). Feuerbach considered the late Schelling and his followers to have annulled this distinction to their own discredit and due to their own lack of "character" (Feuerbach 1973, p. 183).

Between the publication of the Stahl review (in 1835) and the Sengler review (in 1838), Feuerbach published (in 1837) a lengthy work on Leibniz, the most important chapter of which contains a critique of Leibniz's *Theodicy* (Feuerbach 1984). In that work, Leibniz had sought to demonstrate the compatibility of faith and reason in response to arguments to the contrary put forward by Pierre Bayle. The central claim advanced by Feuerbach against Leibniz is that, in doing so (in his most popular work), Leibniz sought to reconcile two "standpoints" that are fundamentally at odds with one another. These Feuerbach calls, respectively, "the theological standpoint", which conceives of God as an intentional agent who stands in external relation to the world, and "the philosophical standpoint", which, by contrast, conceives of individual things as modes of the one, infinite substance. At one point, Feuerbach compares Leibniz's effort to reconcile these two standpoints to the futile attempt made by the astronomer Tycho Brahe to synthesize the Ptolemaic and Copernican models of planetary motion. The argument developed here against Leibniz is structurally similar both of Jacobi's previously mentioned rejection of efforts like Herder's to produce a *geläuterte* Spinozism, as well as the central charge that Feuerbach levels against the Positive Philosophers' allegedly misguided efforts to reconcile Jacobi's personalism with the aims of philosophical science.

Insufficient attention has been paid by historians of philosophy to the fact that the distinction drawn by Feuerbach in the Leibniz book between the philosophical and theological standpoints is modeled after the distinction drawn by Spinoza in his *Ethics* between images (or "*entia imaginationis*") and ideas (or "*entia rationis*").[16] This distinction underlies Spinoza's observation, in the *Tractatus Theologico-Politico*, that the biblical authors "imagined God as a ruler, legislator, king, merciful, just, etc., despite the fact that all the latter are merely attributes of human nature and far removed from the divine nature" (de Spinoza 2007, p. 63). That the same distinction underlies the central thesis of *The Essence of Christianity*, according to which the God of religion is an alienated projection of the human *Gattungswesen* or species-essence, is reflected in Feuerbach's comment that "it is not I, but religion itself, that repudiates and negates the God that is not human, but is an *ens rationis*" (Feuerbach 1973, p. 14). It is at the point that God is conceived as an ens rationis that he ceases to be a personality "*in concreto*" (and thereby also the proper addressee of prayer and object of

worship): "But precisely there, where the personality *in concreto* begins, philosophy is at its end" (Feuerbach 1969c, p. 182).

In *The Essence of Christianity*, Feuerbach uses the expression "God as God" to refer to the impersonal God who is the object of philosophical consciousness. This he distinguishes from the anthropomorphically conceived God of religious faith, who is the object of prayerful supplication, and of whom Christ is believed to be the perfect image. "The humanity of God is his personality", Feuerbach writes. "God is a personal being means: God is a human being" (Feuerbach 1973, p. 256). That Feuerbach's anthropotheism is formulated in direct response to the Positive Philosophers is a point that has too often been overlooked. So, too, has been his charge that, in thus elevating the finite human subject to the status of the Absolute (by conceiving of God as a kind of super-human individual), they had committed *both* a category mistake and an act of conceptual idolatry.

In the preface to the first edition of *The Essence of Christianity* (1841), Feuerbach says that this book contains the elements of his own "philosophy of positive religion or revelation" (Feuerbach 1973, p. 3). In the preface to the second edition (1843), he acknowledges that his aim in developing this philosophy of positive religion was to dismantle both the Hegelian claim for the identity of the content of religious and philosophical consciousness as well as the conception of divine personality advocated by representatives of the Positive Philosophy (Feuerbach 1973, pp. 10–11). He also remarks—after having invoked the names of Jacobi and Schleiermacher—that whoever is unfamiliar with the historical presuppositions and "stages of mediation" (*Vermittlungsstufen*) underlying his arguments lacks the requisite point of entry for making sense of them (Feuerbach 1973, p. 24).

In fact, a central component of Feuerbach's argumentative strategy in *The Essence of Christianity* involves his deployment against Hegel of resources derived from Jacobi and Schleiermacher, though not in ways in of which either would likely have approved. This is most evident in Feuerbach's emphasis on the dramaturgical nature of religion, and on the centrality of feeling and imagination, which had been relegated to the periphery by Hegel.[17] This is also evident, as shall be shown in more detail below, both in Feuerbach's appeal to the empirical facts of religious consciousness and in his insistence that these facts be allowed to speak for themselves rather than being subsumed them within a preconceived theoretical system. Unlike Jacobi and Schleiermacher, however, Feuerbach's "practical-therapeutic purpose" in seeking to explain how anthropomorphic conceptions of the divine arise in the human mind, and to identify the needs that they serve, as well as the processes that give rise to them, is thereby to limit their influence. This is because he had come, in the age of Metternich, to regard this influence as an obstacle to the intellectual coming of age—in Kantian terms, the "*Mündigkeit*"—of the human race and the German people in particular.

As noted previously, Jacobi's affirmation of the existence of a personal God is closely related to his affirmation of the freedom of the human will from natural necessity. Feuerbach agrees but regards this as a merely imaginary and compensatory freedom. Belief in the existence of a personal God, on the account developed by him in his later writings on religion, results from the painful constraints that nature imposes on the finite human subject, including, most fundamentally, the constraint of mortality and the powerful wishes and desires that these constraints arouse in us. Is it too much to suggest that the central place occupied by the concept of "the wish" in Feuerbach's last word on religion, i.e., in his *Theogony*, can itself be traced back to the influence of Jacobi? Consider that Jacobi once wrote, "It is impossible that everything be nature and that there be no freedom for it is impossible that what alone ennobles and elevates man (*truth, goodness, beauty*) be only delusion, deception, lie" (Jacobi 1994, pp. 531–32). What is worth noting is that the word "impossible" here presumably does not mean "logically contradictory" or "inconceivable" but something more like "existentially intolerable". This interpretation is consistent with Sandkaulen's characterization of the rationale underlying Jacobi's *salto mortale* when she writes: "Spinoza demands a revision of our belief in the freedom of our actions that is so radical, a revision to our conception of ourselves as agents and of the lifeworld we inhabit

that is so fundamental, that the prospect of actually putting his conception into practice proves to be entirely unbearable" (Sandkaulen 2023, p. 21).

In light of these considerations, I propose reading the following passage from the chapter in *The Essence of Christianity* on "The Mystery of the Christian Christ or of the Personal God" as Feuerbach's own commentary on the logic of Jacobi's *salto mortale*:

> Desire says: There must be a *personal* God, he can*not not* be. The satisfied heart [replies]: *He exists*. The *guarantee* of his existence lies for the heart in the *necessity* of his existence: the necessity of the satisfaction of the *violence* of the need. It knows no law outside of itself. (Feuerbach 1973, p. 258)

The necessity that motivates the *salto mortale*, on the interpretation proposed here, is not of a logical but rather of a strictly *psychological* nature. To say the same thing in Feuerbach's language, the need that this leap is intended to meet is not a *theoretical* need but a *practical* one. In his later writings, Feuerbach will tend increasingly to identify the need that gives rise to religion with the need to be free from the limitations by which the human drive-to-happiness (*Glückseligkeitstrieb*) is restricted. It is for empirical scientific research (now no longer identified with "speculation") and modern medicine to discover how the restrictions imposed on this drive by nature can be mitigated, if not removed entirely, and for political reform to ameliorate restrictions due instead to historical circumstances subject to modification by the will of a properly educated citizenry (cf. Gooch 2013b).

Feuerbach's primary reason for referring to "the essential standpoint of religion" as "the practical standpoint" is his conviction that the fundamental purpose of religion is not to discover the truth but to secure "the well-being, salvation and blessedness of human beings" (Feuerbach 1973, p. 316). Whereas Hegel thought that the fundamental aspiration of religious consciousness is to relate itself to "a substantial content that is independent and self-subsistent, a truth that is not a matter of opinion and intellectual conceit but which is *objective*" (Hegel 2002, p. 342), Feuerbach's position is that religion appeals instead to the emotions (*das Gemüt*), to the drive for happiness, and especially to the affects of fear and hope (Feuerbach 1973, p. 318). "It is not the absolute as such that is the object and content of religion, but the absolute only *as* it is an object of feeling and imagination—*that* absolute whose essential determination is constituted precisely by this 'as'" (Feuerbach 1969d, p. 220).

What makes the standpoint of religion "practical" in Feuerbach's estimation is that it is determined by subjective human needs. Prayer is the characteristic form of religious activity through which the religious person seeks to satisfy these needs. The religious representation of God as an omnipotent and merciful being who responds to prayer is, on this account, determined by the needs that the activity of prayer is intended to meet rather than by a theoretical motivation to discover an adequate idea of the Absolute. Whereas Hegel "finds the quintessence of religion only in the *compendium* of *dogmatics*", Feuerbach claims to have discovered it instead "already in the *sample act* of prayer" (Feuerbach 1970b, p. 231).

A text of Feuerbach's that is especially relevant for evaluating the "break with speculation" that he sought to make in the early 1840s is the preface to the second edition of *The Essence of Christianity*, published the same year (1843) as his *Principles for a Philosophy of the Future*. In this preface, Feuerbach repudiates "the absolute, the immaterial speculation that is satisfied with itself—the speculation that creates its own content (*Stoff*) from itself", emphasizing that he had sought instead to solve "the riddle of the Christian religion" through "an empirical or historical-philosophical analysis" (Feuerbach 1973, p. 14). In explicit contrast to the Hegelian ideal of presuppositionless science, and echoing Jacobi's critique of the rationalist paradigm of scientific demonstration, Feuerbach emphasizes that the claims advanced by him in this book are "only the conclusions, the inferences from premises that are not themselves thoughts, but rather objective, either living or historical, facts" (Feuerbach 1973). It is striking that, in explicit and direct connection with his repudiation of "speculation", Feuerbach should invoke Jacobi by directly quoting a famous passage from the Spinoza Letters. He does this when he emphasizes that, in contrast to

speculative philosophy, his theoretical aim had not been to invent (*erfinden*) but rather to discover (*entdecken*) the essence of Christianity and, in so doing, "to reveal existence" (*Dasein zu enthüllen*). This is precisely what Jacobi himself had identified in the Spinoza Letters as the principal task of scientific inquiry.

In Rawidowicz's monumental survey of Feuerbach's entire corpus, and of the relation of his ideas to those of various predecessors and contemporaries (originally published in 1931), he noted a more sympathetic tone in comments about Jacobi found in Feuerbach's later writings compared with the reserved tone of remarks made prior to his "break with speculation" (Rawidowicz [1931] 1964, p. 262). He attributed this more sympathetic tone to a number of points of agreement between Jacobi's position and the "philosophy of the future" that Feuerbach first sought to sketch out in 1843. These include a common appeal to immediate knowledge that is expressed both in Jacobi's doctrine of faith (*Glaube*) and in Feuerbach's defense of sense certainty against Hegel's famous critique of it in the *Phenomenology*; a common commitment to "realism" understood as the view that our perceptions involve a direct awareness of extra-mental material objects and are not mere representations of such objects; a corresponding rejection of the speculative claim for the identify of thought and being; and a common emphasis on the I–Thou relationship or the inter-subjective structure of human subjectivity (in contrast to the Cartesian cogito). In the foregoing, I have sought to explain why these points of agreement, far from being coincidental, are instead reflective of Feuerbach's longstanding constructive engagement with Jacobi, whose influence on nineteenth-century German philosophy extends further than has generally been recognized.

**Funding:** This research received no external funding.

**Data Availability Statement:** No new data were created or analyzed in this study. Data sharing is not applicable to this article.

**Conflicts of Interest:** The author declares no conflict of interest.

## Notes

[1]  They were so taken by Friedrich Engels in his book, *Ludwig Feuerbach und die Ausgang der klassichen deutschen Philosophie* (Engels 1886), to which the title of this article alludes (Engels (1886)).

[2]  This and all other translations from the German are by the author (T.G.) unless otherwise noted.

[3]  Further insight into Feuerbach's reasons for making such a break at this time are contained in a handwritten manuscript discovered in the Feuerbach *Nachlass* and published by Ascheri Carlo with extensive commentary in Ascheri (1969). Relevant factors include a change in the political climate reflected in the closure of the *Halle Annals* by the Prussian censor, as well as a police search of Feuerbach's personal residence in Bruckberg.

[4]  An important exception in this regard is Rawidowicz ([1931] 1964), a study that remains unsurpassed for its comprehensiveness and includes a chapter on Feuerbach's relation to Jacobi (pp. 258–65), to which further reference is made below.

[5]  The term "*Vormärz*" refers to the period in German and Austro-Hungarian history that preceded the failed revolution that broke out in March of 1848.

[6]  This and subsequent Jacobi quotations are from the 1785 edition of "Concerning the Doctrine of Spinoza" unless otherwise indicated.

[7]  This passage is from the 1789 edition of the Spinoza Letters.

[8]  This passage, and those cited in the next paragraph, are from Supplement 2 to Jacobi's open letter to Fichte, first published in 1799.

[9]  This passage is from the preface to the 1815 edition of Jacobi's *David Hume and Faith*.

[10] As Merold Westphal observes in his own foreword to the translation of Hegel's "Foreword to Hinrich's *Religion in Its Inner Relation to Science*" (Hegel 2002, p. 334), Hegel uses the term "secular" in this context as a synonym for the understanding (*Verstand*) in contrast to reason (*Vernunft*). The translation of Hegel's foreword found on pp. 337–53 of this volume is by A. V. Miller.

[11] Hegel (1988) is an abridgment of a three-volume edition of Hegel's lectures on the philosophy of religion based on the critical edition of these lectures found in Hegel (Hegel 1983–1985).

[12] Hegel attributes Spinoza's inability here to the grip on him of a certain "Oriental intuition" of the oneness of God due to his having been a Jew "by descent". What he finds missing from Spinoza's philosophy is "the Occidental principle of individuality" exemplified by Leibniz's theory of monads (Hegel 1991, p. 226 [§ 151 Z]).

13   See, in this context, Hegel's claim in the introduction to his early essay on *Faith and Knowledge* that Kant, Jacobi, and Fichte, who are said here to share the same "fundamental principle [of] the absoluteness of finitude", had together "raised . . . the culture (*Kultur*) of reflection . . . to a system" (Hegel 1977, pp. 62 and 64).

14   The first of these is Feuerbach's review of the first two volumes of Friedrich Julius Stahl's *Philosophy of Right from a Historical Perspective* (1830, 1833), published in the *Berlin Annals* in 1835 (Feuerbach 1969a), and the second is his review of Jacob Sengler's *The Essence and Significance of Speculative Philosophy and Theology in the Present Age* (1837), published in the *Halle Annals* in 1838 under the title, "Toward a Critique of the 'Positive Philosophy'" (Feuerbach 1969c).

15   Bowie translates "*Glaube*" here as "belief".

16   Cf. the appendix to Part 1, Proposition 36, as well as Book 2, Propositions 40–49.

17   At the outset of *The Essence of Religion* (1846), Feuerbach explicitly identifies the *Abhängigkeitsgefühl* or "feeling of dependence" as the "ground" of religion while also identifying nature as the original "object" of this feeling, and hence of religion (Feuerbach 1971, p. 4). In a reply to a reviewer entitled "Zur Beurteilung des Schrifts 'Das Wesen des Christentums'" (1842), he suggests that the difference between his own position and Hegel's is most clearly evident in their respective attitudes toward Schleiermacher before going on to remark that Hegel failed to penetrate "the essence of religion . . . because he as an abstract thinker was not able to penetrate the essence of feeling" (Feuerbach 1970b, p. 230). Jacobi's influence is less readily detectable but, I believe, no less important.

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
