# Peer review of "Faith, Knowledge, and the Ausgang of Classical German Philosophy: Jacobi, Hegel, Feuerbach"

_religions, doi:10.3390/rel15050618_

Round 1

Reviewer 1 Report

Comments and Suggestions for Authors

I like this paper a great deal: the summaries of Jacobi and of Hegel are very nicely done, and the discussion of Feuerbach is illuminating. I only wish there were more substance to the discussion of Feuerbach's entanglement with Jacobi: given all the build-up, it seems to end too soon.

Also, the paper is full of spelling errors, grammatical problems, and so on. I trust those will be corrected in the editing process.

But generally, great job.

Comments on the Quality of English Language

The English is very natural but there are many, many spelling errors and a few places where the grammar is nonsensical. Make sure this paper gets a thorough copyediting.

Author Response

Thanks to the reviewer for taking the time to read through my article manuscript and for the helpful feedback, especially with regard to the concluding section on Feuerbach, which I agree ends somewhat abruptly and is a little anti-climactic. This is partly due to the false assumption I had been working under that there was an 8,000-word limit. On submitting the manuscript shortly before the deadline, I realized the actual limit was 10,000. This will allow me to add some additional material to flesh out the Feuerbach section prior to submitting the revised manuscript.

I apologize for the number of typographical errors and a couple of grammatical issues. I believe these have now all been identified and corrected.

Reviewer 2 Report

Comments and Suggestions for Authors

Argumentation and structure of article "Faith, Knowledge, and the Ausgang of
Classical German Philo
sophy" are convincingly presented.

The author, does not merely prove to be knowledgeable on the subject matter but also of possessing a deep grasp of the question discussed.

More than that, the paper is written in academic English but it is also clear to follow.

One minor suggestion is to reformulate the Introduction in such a way as to immediately present the aim of the paper.

Author Response

Thanks to the reviewer for taking the time to read and comment on the draft of my article. I apologize for the number of typos contained in the manuscript I submitted. These have been corrected.

I have followed the reviewers suggestion to reformulate the opening paragraphs so as to present the aim of the article concisely at the beginning.

Reviewer 3 Report

Comments and Suggestions for Authors

This is a very interesting article that sheds light on the development of philosophy of religion in the wake of German idealism in a very intersting way. I have some short comments the author can include in a revision of his article. There are some typos and some questions concerning language that I highlighted in the pdf. Thank you for the interesting paper. 

Author Response

Thank you for your helpful feedback. I apologize for the number of typographical errors in my manuscript, which have now been corrected.

Prior to submitting my revisions I will add several sentences to the article in response to the reviewer's suggestions.

Reviewer 4 Report

Comments and Suggestions for Authors

One quibble I have is how the author never directly engages Schelling’s “positive philosophy”. While frequently named, it is never presented or analyzed, even in passing. An omission which most won’t appreciate, but which those familiar with Schelling’s work will. Specifically, the Berlin Lectures 1841/42, which Feuerbach attended, and wherein Schelling introduced his “existential system” of philosophy. Thus, when the author writes “I attempt to show how Feuerbach’s critique of the Hegelian claim for the identity of religious and philosophical truth, as well as his critique of the so-called “Positive Philosophy” inspired by the late Schelling, were influenced both by Jacobi and by Spinoza in crucial ways that have yet to receive sufficient consideration” (p.3), there is quite a large missing piece in this genealogical puzzle. Again, the author writes of the aftereffects of Feuerbach’s writing, including “Existenzphilosophie with its characteristic concerns for facticity, temporality, corporeality, and finitude” (p.3), all extensively introduced and developed by Schelling in these lectures Feuerbach attended. As a Schelling scholar this is an omission in need of addressing. If the author is so inclined, I would recommend F. W. J. Schelling, Philosophie der Offenbarung 1841/42, ed. Manfred Frank (Frankfurt am Main: Suhrkamp, 1977) and the translators introduction in The Grounding of Positive Philosophy: The Berlin Lectures (Suny Series in Contemporary Continental Philosophy, 2007).
